# Dehydration Treatment Improves *Ulmus glabra* Dormant Bud Regeneration from Cryostorage

**Sakari Välimäki \***, **Mari Rusanen and Tuija Aronen**

Natural Resources Institute Finland (Luke), FI-57200 Savonlinna, Finland
* Correspondence: sakari.valimaki@luke.fi; Tel.: +358-29-532-2127

**Abstract:** The conservation of genetic resources in cryocollections requires reliable protocols for the cryopreservation and the regeneration of the preserved material. With *Ulmus glabra*, the regeneration of thawed buds by in vitro organogenesis has suffered from low shoot growth and high contamination rates. The dehydration of the buds before cryopreservation improved the shoot growth rate and ameliorated the contamination rate of in vitro cultures initiated from thawed buds, although the degree of success varied depending on the donor tree.

**Keywords:** wych elm; vegetative propagation; micropropagation; tissue culture; cryopreservation; genetic resources

## 1. Introduction

Cryoconservation methods have been developed for supplementing the protection of genetic diversity of *Ulmus* species, which are threatened by Dutch elm disease [1–6]. Dormant winter buds are relatively easy and cost-efficient to cryopreserve, as they do not require pre-existing in vitro cultures or extensive pre-treatment [7]. However, the cryopreservation success varies depending on the cold hardiness of the material during bud collection [2,8,9]. The multiplication of shoots in vitro allows generation of multiple plants from a small initial explant. For successful in vitro culture initiation, the buds need to be surface sterilized after thawing as the material is collected from outdoors and frozen without disinfecting pre-treatments.

In Finland, efforts have been taken to put together cryocollections of the native elm species, *Ulmus laevis* Pall. and *Ulmus glabra* Huds. [6]. However, the regeneration after thawing and reliable surface sterilization of *U. glabra* have proven to be challenging. Compared to *U. laevis* buds, *U. glabra* buds are larger and their bud scales are hairier, which could result in contamination sources being harder to reach by sterilizing agents.

Cryopreservation protocols aim to reduce the damage inflicted on the samples by the formation of ice crystals during either the freezing or the thawing of the samples. The buds are pre-cooled slowly in order to transfer water from the cells into extracellular space [10]. In addition, the moisture content can be reduced by dehydration treatments [11,12]. With dehydration, the moisture content of the samples must not be allowed to decrease too much, as excess drying also damages the samples. In many protocols the samples are dehydrated to between 25% and 30% [13–16]. However, the optimal moisture content varies among different species [15] and can be as high as 40% [17].

The aim of the present work was to study the efficacy of dehydration treatment in improving *U. glabra* bud regeneration after thawing from cryostorage.

## 2. Materials and Methods

The twigs (ca. 10–15 cm) with varying numbers of dormant *U. glabra* buds (Figure 1a) were collected from five donor trees (clone numbers 106 02, 107 01, 208 05, 210 07, 323 03) on 16 February 2022 from Preitilä gene reserve collection 184 in Southwest Finland. During

the sampling, the temperature varied around +2 °C (±0.5 °C) and the mean temperature of the month was −2.2 °C (min −18.1 °C, max +5.2 °C), measured at the closest weather station 13 km from the collection site. The collected twigs were transported by car to the laboratory the same day and stored in a cold room (+2 °C) overnight. The buds were cut from the twigs the next day and placed into 1.8 mL cryovials (Sarstedt) with 7–12 buds per vial and kept on ice overnight. The buds were cut from the twigs so that a wood piece about half of the length of the bud remained attached to the bud for easier handling (Figure 1b). The samples were frozen using a programmable freezer (Planer Kryo 10) 0.17 °C min$^{-1}$ until the samples reached −38 °C [18]. The vials were directly transferred from the freezer into liquid phase nitrogen in the storage tanks.

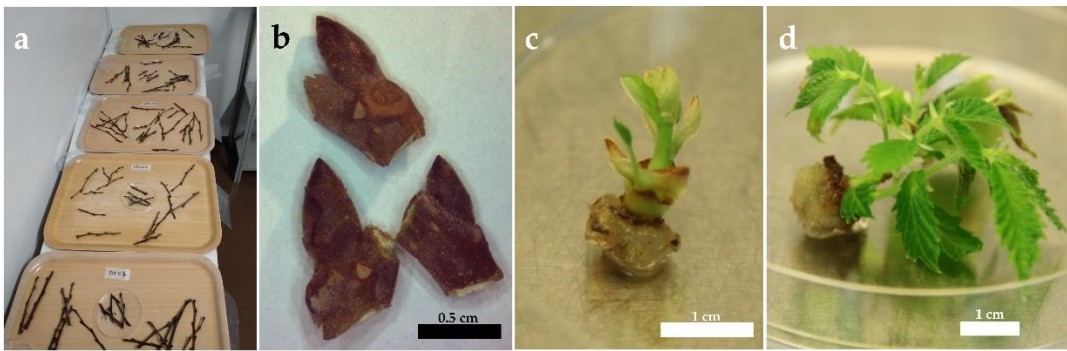

**Figure 1.** (**a**) Twigs in dehydration treatment. (**b**) Buds cut for cryopreservation. (**c**) A successful shoot initiation from a dehydrated bud of the donor tree 107 01, 10 days after thawing and preparation for initiation, i.e., surface sterilization and excision of outer bud scales and most of the wood material at the stem. (**d**) An established in vitro culture from a bud from the donor tree 106 02, 39 days after initiation and 15 days after transfer from initiation to multiplication medium.

For dehydration treatment, the moisture content of the twigs was determined by weighing twigs from all five donor trees before and after drying them in an oven (+90 °C) until the weight of the sample no longer decreased. The initial moisture content of the twigs was determined to be on average 52%. For dehydration, the twigs were placed on trays in a cold room in darkness (−5 °C, ca. 60% relative moisture) as described by Tanner et al. [15] (Figure 1a). The moisture content of the twigs was monitored by weighings and the dehydration was discontinued after 19 days when the twigs reached, on average, 32% moisture. The dehydrated buds were prepared for cryopreservation and frozen in the same manner as non-dehydrated buds.

Before culture initiation, cryovials were thawed in water bath (+38 °C, 2 min) and then kept on ice for 2 min [6,18]. One vial for each tree in both treatments were thawed, except for trees 107 01 and 208 05, for which two vials were thawed per treatment. Surface sterilization was done with 20 g L$^{-1}$ sodium dichloroisocyanurate (NaDCC) in agitation as recommended by Fenning et al. [19], except sterilization time was increased to 1 h, and additionally the buds were kept in 70% EtOH for 15 min. The buds were rinsed with sterilized water three times and left in water until preparation (up to 1.5 h). The preparation was done by excising the outer bud scales and shaping the stem into a small wedge [6]. The cultures were initiated onto Driver and Kuniyuki Walnut (DKW) [20] initiation media made using basal salt mixture (Sigma Aldrich, St. Louis, MO, USA) and supplemented with 30 g L$^{-1}$ sucrose and 1 g L$^{-1}$ myo-inositol, and solidified with 6 g L$^{-1}$ Plantagar S1000 (B&V, Parma, Italy) [6,19]. The plant growth regulators were 0.1 mg L$^{-1}$ gibberellic acid (GA) 4/7 (Duchefa Biochemie, Haarlem, The Netherlands) and 0.5 mg L$^{-1}$ 6-benzylaminopurine (BA) (Sigma Aldrich, Beijing, China). The sprouted buds (Figure 1c) were moved to the fresh medium of the same composition but without GA (multiplication medium, Figure 1d). The buds were transferred after they sprouted, mostly around 10 days from initiation, although there was variation. When initiation success was evaluated, only cultures with growing shoots and without any contaminations (Figure 1c) were considered

successful in terms of shoot production. The results were analysed with IBM SPSS Statistics 28 using logistic regression with the donor tree and the dehydration status as covariates. A *p*-value < 0.05 was considered statistically significant.

### 3. Results

Dehydration improved *U. glabra* buds' regeneration after thawing but there was variation among the donor trees (Figure 2). According to the logistic regression model, dehydration status and donor tree contributed to shoot sprouting and together increased the correctly predicted cases from 70.2% to 80.2% (Table 1). Neither covariate improved the prediction on its own. In this experiment, for all five trees, from each cryovial of dehydrated buds (7–12 buds), at least one vigorous in vitro culture was established.

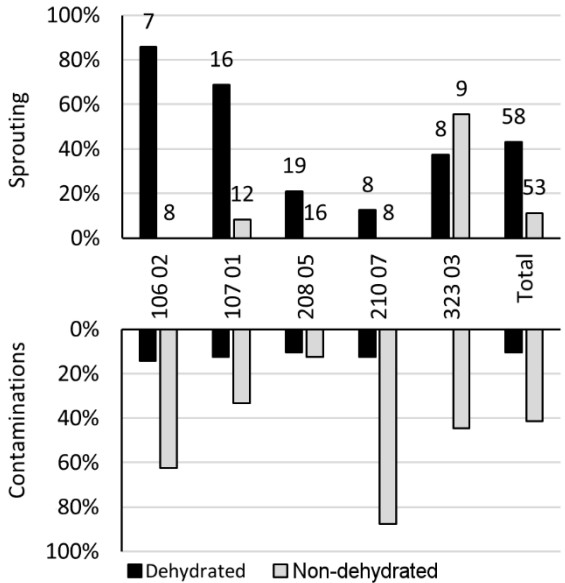

**Figure 2.** The shoot growth rate (**upper chart**) and contamination rate (**lower chart**) of dehydrated and non-dehydrated cryopreserved buds after thawing and initiation of micropropagation. The data were collected during the month after initiation, for both shoot growth and contaminations. For sprouting to be marked as successful, it also had to be without contaminations. The total number of buds thawed and used from each donor tree for the experiment is given at the top of the bars.

**Table 1.** A logistic regression model of shoot production after thawing. Dehydration status and donor tree were used as covariates.

| Model, $log(p/1 − p)$ | Variable | *p*-Value | Odds Ratio (95% CI) | Donor Tree | % of Cases Predicted Correctly by Model |
|---|---|---|---|---|---|
| $log(p/1 − p) = −1.539 + 2.181d − 0.113t_1 − 2.099t_2 − 2.722t_3 + 0.370t_4$ | | | | | 80.2 |
| | dehydration status | <0.001 | 8.851 (2.879–27.209) | | |
| | Tree | 0.004 | Reference | 106 02 | |
| | | 0.88 | 0.893 (0.207–3.852) | 107 01 | |
| | | 0.012 | 0.123 (0.024–0.629) | 208 05 | |
| | | 0.027 | 0.066 (0.006–0.729) | 210 07 | |
| | | 0.654 | 1.447 (0.288–7.285) | 323 03 | |

Dehydration also decreased contamination rates (Figure 2). According to the logistic regression model (Table 2), contamination rate was a combined effect of both dehydration status and donor tree. Including both in the model improved the correctly predicted cases from 74.8% to 82.0%. The origin of the contamination was fungal in nine non-dehydrated buds and in one dehydrated bud, and bacterial in thirteen non-dehydrated and in five dehydrated buds.

**Table 2.** A logistic regression model of contamination rates after initiation of micropropagation. Dehydration status and donor tree were used as covariates.

| Model, $log(p/1-p)$ | Variable | $p$-Value | Odds Ratio (95% CI) | Donor Tree | % of Cases Predicted Correctly by Model |
|---|---|---|---|---|---|
| $log(p/1-p) = 0.400 - 1.966d - 0.846t_1 - 1.738t_2 + 0.583t_3 - 0.909t_4$ | | | | | 82.0 |
| | dehydration status | <0.001 | 0.140 (0.047–0.414) | | |
| | Tree | 0.045 | Reference | 106 02 | |
| | | 0.270 | 0.429 (0.095–1.930) | 107 01 | |
| | | 0.032 | 0.176 (0.036–0.857) | 208 05 | |
| | | 0.475 | 1.791 (0.362–8.856) | 210 07 | |
| | | 0.284 | 0.403 (0.076–2.125) | 323 03 | |

## 4. Discussion

The regeneration of *U. glabra* buds from cryostorage through micropropagation is difficult based on our previous study where the success rate was only around 10% [6]. This is in line with our current results as cryopreservation without dehydration yielded the recovery rate of 11%. The success rate of 43% achieved in the present study with dehydration is a clear improvement. Successful regeneration of *U. glabra* from cryopreservation has been reported without dehydration but with using smaller explants and with micrografting (35.8% with three genotypes) [3], at a success rate similar to our current results. With *U. laevis* we have previously achieved regeneration rate of 64% without dehydration [6], highlighting the difference between the species. Very high recovery rates can be achieved with cryopreservation of in vitro material using various pre-treatments, as reported by Uchendu et al. for *U. americana* [4]. However, the costs of the cryopreservation of in vitro material are higher [21].

According to Tanner et al. [15], dehydration before cryopreservation is necessary for all but the most cold-hardened plant material. Even though *U. glabra* in Finland go through sub-zero temperatures during the winter, dehydration treatment improved the regeneration of the explants. In addition to the dehydration, the extra time in constant sub-zero temperature may have improved the cold hardiness of the material as suggested by Forsline et al. [13]. To rule out reduced deacclimitization during transport, the material was delivered by car the same day after collection, instead of mailing it with cold packs as done by Välimäki et al. [3]. Despite this, the control buds that were cryopreserved without dehydration regenerated poorly. The temperature during and a few days before bud collection was above zero, but before that there was a period of multiple weeks of consistent night frosts. However, the results could be improved by more precise timing of the collection.

Surprisingly, the dehydrated buds had fewer contaminations than non-dehydrated control buds. This could be due to the sterilizing agent penetrating deeper because of the lower moisture content of the dehydrated buds. Previously, sterilization of dehydrated buds with 0.2% $HgCl_2$ for 10 min was tested unsuccessfully (data not shown). With NaDCC the sterilization time was 1 h, which gives the sterilizing agent more time to penetrate the buds and reach the contamination sources. This is further supported by rehydration after thawing in water overnight having been tested with buds from two donor trees, which resulted in a higher contamination rate (data not shown).

The improved protocol enables the meaningful use of cryopreservation of *U. glabra* as a backup method for genetic resources conservation. Dehydration is an extra step to the protocol, but it only requires the initial weighing and the weekly monitoring of the moisture content, which are not extensively laborious. The total successful regeneration rate of 43.1% achieved in the present study would ensure regeneration of at least one bud from a vial of seven buds at the 95% confidence level according to calculations by Volk et al. [22]. As the successful initiation can be further multiplied through micropropagation, even recovery of a single bud can be sufficient for the vegetative propagation of a large number

of cuttings. Still, even with dehydration, the cryocollection design is complicated by the high variation between donor trees. The lowest success was with the donor tree 210 07, for which only one out of eight initiations was successful. Based on this result, 23 buds should be cryopreserved to have at least one successful initiation at the confidence level of 95%, if calculated as $p$(failed initiation)$^n$ = $p$(all initiations failed), where n is the number of initiations. For more precise estimates, the protocol needs to be tested and validated with wider material from more trees, collected through different years. Moreover, dehydration of the buds to more optimal moisture content than tested now (32%) could improve the results further. Nevertheless, our approach is to store a high number of genotypes, and it is not detrimental to lose a few that prove difficult to handle in vitro. The goal is to preserve the genetic diversity of the elms in Finland, rather than specific high-value genotypes. As a conclusion, the dehydration step will be included in the protocol for establishing a cryocollection for *U. glabra* in Finland.

**Author Contributions:** Conceptualization, S.V.; methodology, S.V.; formal analysis, S.V.; investigation, S.V.; resources, M.R. and T.A.; data curation, S.V.; writing—original draft preparation, S.V.; writing—review and editing, S.V., M.R. and T.A.; visualization, S.V.; supervision, T.A.; project administration, M.R.; funding acquisition, S.V., M.R. and T.A. All authors have read and agreed to the published version of the manuscript.

**Funding:** This research was supported by the Ministry of Agriculture and Forestry of Finland under the MEKANEN project (VN/7347/2020).

**Data Availability Statement:** The data presented in this study are available on request from the corresponding author.

**Acknowledgments:** We wish to thank the technical staff at Luke Finland involved in bud collection, laboratory, and greenhouse work.

**Conflicts of Interest:** The authors declare no conflict of interest. The funders had no role in the design of the study; in the collection, analyses, or interpretation of data; in the writing of the manuscript; or in the decision to publish the results.

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
