# Peer review of "Dehydration Treatment Improves Ulmus glabra Dormant Bud Regeneration from Cryostorage"

_forests, doi:10.3390/f13111923_

Round 1

Reviewer 1 Report

This communication presents new interesting ideas for cryopreservation of elm dormant buds. This type of article could be inspiring in the field dormant bud cryopreservation.

I have only small comments regarding some facts in the text. I think authors designed and presented very nice work.

1 “Desiccation” suggest to replace with “dehydration” in the whole article. In general, desiccation is used at higher levels of dehydration stress, well below 32%. Also in cryopreservation studies, dehydration is used rather in this range.

49-51 “The buds were cut from the twigs so that a wood piece about half of the length of the bud remained attached to the bud for easier handling.“ I understand that sentence, but for the quality of this work, I think it should be better explained. If it was a uni-nodal cutting or if you use only a part of the bud adjacent stem. If possible, you could add a picture.

52-53 “The vials were directly transferred from the freezer into liquid nitrogen tanks.” You should better explain if you plunged the vials directly in liquid nitrogen or if you used nitrogen vapours cryostorage system. It affects the final freezing rate much. You used a protocol based on your previous experience. I understand it is not the topic of this article, but it would be interesting if you could state why you did not use an annealing/equilibration phase (often used in dormant bud cryopreservation) before transferring to the liquid nitrogen tank. You also used very fast warming of vials (68), although as you cited Harvengt et al. 2004 [1], he used slower warming rates for elm (30 min complete thawing in the air).

60 You used dehydration to 32% moisture, but at 38-39 you stated: “The most suitable moisture content varies among different species [11,12], but in many protocols the samples are desiccated to between 25% and 30% [13-15].” Why you used the higher level of dehydration than recommended? Have you done any previous experiments with it?

69 “dichloroisosyanurate” should be dichloroisocyanurate

86 “80.1%” should be 80.2., according the table 1.

92-93 “The number of buds initiated from each donor tree is given at the top of the bars.” The word “initiated” is inappropriate - it evokes the number of buds sprouting. Should be something as used/ treated.

114 -116 “In addition to the desiccation, the extra time in constant sub-zero temperature may have improved the cold hardiness of the material as suggested by Forsline et al. [14].” As you mentioned, you collected the samples on 16 February 2022 and stated the average and max/min temperatures of the month. I cannot see the sampling temperature from this - in the future study, I would recommend doing the sampling at the end of the freezing period and ideally keeping the temperature below zero (that -5 °C dehydration temperature).

138 “Based on this result, 23 buds should be cryopreserved to have at least one successful initiation at the confidence level 139 of 95%” Just a note on a practical cryoconservation strategy. Although there is no consensus on the minimum number of stored samples, it is worth considering relying on only one regenerable sample.

Author Response

We wish to thank Reviewer 1 for their valuable comments to improve our manuscript. Please see the attachment for the point-by-point responses.

In addition to the modification requested by the Reviewers, some grammatical errors were corrected and the reference list corrected.

Reviewer 2 Report

The paper presents the result of the research work on cryopreservation of dormant buds of Ulmus glabra, an important tree species. Ulmus species worldwide are threatened by several deadly pathogens including Dutch elm disease mentioned by the authors, hence the importance of the long-term conservation of their genetic resources. Besides common seed conservation methods, conservation of vegetative organs, such as shoot tips or dormant buds, is needed to preserve unique genotypes, for example, with increased disease tolerance or enhanced timber traits. However, woody species are extremely difficult to work with due to specific requirements for the media as well as low propagation rate, and the attempts  to cryopreserve dormant buds of forest trees are limited to very few studies with varied success. From this viewpoint, the results presented here are interesting, novel and worth to be published in Forests. That been said, there are several comments to address during the revision. I hope that they would help to improve further the manuscript which is already well-written.

1.     As I’ve mentioned, there are indeed very few studies in cryopreservation of Ulmus, and all of them worth to be cited at least in the Introduction. Consider including the works by Uchendu et al. on Ulmus Americana (see here https://onlinelibrary.wiley.com/doi/abs/10.1111/jpi.12094)

 and by Paques et al. on three Ulmus species from France (attached)

2.     Line 24. Before the sentence “For successful culture initiation..” it is better to explain why it is necessary to use in vitro culture for regrowth of dormant buds.

3.     Line 34. Better to say “pre-cooled” instead of “pre-frozen”

4.     Line 35, desiccation is just one of the functions of cryoprotectants. Consider avoiding the discussion of cryoprotectants in this paper, otherwise you will need to go deeper into the mechanism of the CPA-induced cryotolerance. You don’t use them anyway.

5.     Materials and methods are sometimes not clear and require additional  details. Line 45 – please mention the geographic region where the material was collected.

6.     What was the average size of the twigs (they seem to be different in the photo in Fig 1a), how many buds per a twig (roughly if not recorded)? What was the average size of the extracted buds?

7.     Line 49. It is said that 7-12 buds were used per vial. But how many buds in total were used for each tree? How many vials you had per one tree in total? This part is not clear from the text.

8.     Line 72 “until preparation onto initiation media”. Do you mean “until placing them onto initiation medium”? how long, roughly, was this period – several hours or several minutes?

9.     Line 73. Did the media contain a carbohydrate source? Please mention the chemical and concentration, if relevant.

10.  Please also mention the source (manufacturer) of medium mixture and growth regulators.

11.  Line 75, “After the buds sprouted..” – how long was this period? Was it different for different trees?

12.  Line 76. The multiplication medium is the same as initiation medium but without GA, correct? In this case, you may write that “The sprouted buds (Figure 1b) were moved to the fresh medium of the same composition but without GA (multiplication medium).”

13.  Figure 1. Legend. “A successful initiation from a desiccated bud of the donor tree 107 01, 10 days after thawing and preparation”. What you mean by “preparation”? Also, it may be better to say “successful shoot initiation”.

14.  Line 66. Words “in shoot multiplication” after tree number could be removed, you have it mentioned later in the same sentence.

15.  Results. Fig. 2. “Control” label in Fig 2 is confusing (one may initially think that these are non-cryopreserved buds). Consider changing to “non-desiccated”.

16.  Line 92. “The number of buds initiated from each donor tree is given at the top of the bars…”  So, for example, the first bar in Fig 2 top graph indicates that 7 buds from the tree 10602 survived cryopreservation and they represented 80% of all buds taken to cryopreservation? This is not clear at the first glance and this is why it is important to indicate the number of buds processed for each tree.

17.  Please include in Fig 2 legend the period when the data were counted (days or weeks after rewarming).

18.  It would be also good to know the dynamics of water content change during desiccation (since you’ve monitored it anyway).

19.  Discussion. I would appreciate a discussion on how the survival rate achieved in your study is comparable (or better) than the survival achieved by other researchers who worked with elm dormant buds and shoot tips in vitro. In vitro culture may be laborious but it may provide better regrowth rates and hence will be more efficient, or not? Also, it would be good to discuss the optimum and/or safe ranges of water content for successful cryopreservation of different tree species based on yours and other researches on elm and other tree species. Is the safe water content for cryopreservation of dormant buds similar for different tree species?

At last, not a critical remark but rather a suggestion or as ideas for the future: it would be interesting how the water content of buds was different (or not different) from the water content of the twigs. Usually, buds are required to be measured separately for water content.

Author Response

We wish to thank Reviewer 2 for their valuable imput and comments to improve our manuscript. Please see the attachment for the point-by-point responses.

In addition to the modification requested by the Reviewers, some grammatical errors were corrected and the reference list corrected.
